# Linear and Nonlinear Modes and Data Signatures in Dynamic Systems Biology Models

**Joseph DiStefano III**

Departments of Computer Science and Medicine, University of California at Los Angeles (UCLA), Los Angeles, CA 90095-1596, USA; joed@cs.ucla.edu

**Abstract:** The particulars of stimulus–response experiments performed on dynamic biosystems clearly limit what one can learn and validate about their structural interconnectivity (topology), even when collected kinetic output data are perfect (noise-free). As always, available access ports and other data limitations rule. For linear systems, exponential *modes*, *visible* and *hidden*, play an important role in understanding data limitations, embodied in what we call dynamical *signatures* in the data. We show here how to circumscribe and analyze modal response data in compartmentalizing model structures—so that modal analysis can be used constructively in systems biology mechanistic model building—for some nonlinear (NL) as well as linear biosystems. We do this by developing and exploiting the modal basis for dynamical signatures in hypothetical (perfect) input–output (I-O) data associated with a (mechanistic) structural model—one that includes inputs and outputs explicitly. The methodology establishes model dimensionality (size and complexity) from particular I-O datasets; helps select among multiple *candidate* models (model distinguishability); helps in designing new I-O experiments to extract "hidden" structure; and helps to simplify (reduce) models to their essentials. These modal analysis tools are introduced to NL enzyme-regulated and protein–protein interaction biosystems via nonlinear normal mode (NNM) and quasi-steady state approximation (QSSA) analyses and unified with linear models on invariant 2-dimensional manifolds in phase space, with properties similarly informative about their dominant dynamical properties. Some automation of these highly technical aspects of biomodeling is also introduced.

**Keywords:** mechanistic model; model distinguishability; quasi-steady state approximation; QSSA; hidden modes; visible modes; dynamical signatures; candidate model; systems pharmacology; epidemiology; minimal model; hidden Markov model; conservation constraints

## 1. Introduction

Over the last few decades, interest and progress in mechanistic math modeling has burgeoned in the realm of dynamic systems biology [1,2], systems pharmacology [3] and epidemiology [4]. The subject literature has grown substantially, with much greater attention being paid to quantitative modeling methodology and applications, particularly in the university life science and pharmaceutical communities. Hindsight suggests that this focused diligence was to be expected. The development of vastly superior molecular and cell biology measurement tools and the numerous biological discoveries emanating from the explosive growth of data provided by these tools was undoubtedly a major if not the impetus. The common understanding is that explicit quantitative models transform data into useful constructs, ideas and discoveries. And the literature is growing with papers on modeling methodologies specialized for molecular and cellular level modeling and textbooks that are consolidating them, e.g., [5–10].

Multicompartmental modeling, described in some detail in Sections 4 and 5 below, has been a prominent methodology in the quantitative physiology and pharmacology literature for over half a century, yet it remains an area underappreciated by systems biology modelers. Indeed, the subject of compartmental modeling occupies few if any pages in

modern texts on systems biology, as for most of the references above. An overarching goal in this paper is, first, to make the case that this oversight may be hampering progress in the field—at least in the sense that readily available and effective tools for mechanistic modeling are being neglected. To accomplish this, I summarize the methodology and show the ease with which dynamic systems input–output (I-O) data can be exploited in structural, mechanistic model building, first using linear systems analysis modal concepts merged with linear compartmental modeling. This is developed and exemplified for linear system compartmental models. And I show how dynamical data signatures are transformed into minimal model structures and also how to analyze particular candidate model structures (model distinguishability). This is key to unraveling structure from data and designing new input–output (I-O) experiments that reveal more. Then, I show how to extend and unify these notions to nonlinear dynamic systems biology modeling by merging concepts in structural mechanics with enzyme kinetic modeling methodology and extending them to modeling and understanding biochemical kinetics. First, some theories and examples.

## 2. Linear Modes

Modal concepts are classical in linear system theory, e.g., see reference [11]. Consider the general class of linear time-invariant ordinary differential equation (ODE) models, with inputs and outputs specified integral to model $M$, as in Figure 1 and Equation (1):

$$\frac{d\boldsymbol{q}(t)}{dt} = K\boldsymbol{q}(t) + B\boldsymbol{u}(t), \quad \boldsymbol{y}(t) = C\boldsymbol{q}(t), \quad t \geq 0 \tag{1}$$

Symbols $\boldsymbol{q}$, $\boldsymbol{u}$ and $\boldsymbol{y}$ are $n$-, $r$- and $m$-vectors of time-varying state variables (system), inputs and outputs, respectively. $K$, $B$ and $C$ are $n$ by $n$ system, $n$ by $r$ input and $m$ by $n$ output matrices of constants, respectively. Multicompartmental (MC) models, defined graphically in Section 4 below, are a special case, with $K \equiv [k_{ij}]$ having the properties $k_{ij} \geq 0$ for $i \neq j$ and $\sum_{i=0}^{n} k_{ij} = 0$, $j = 1, \ldots, n$ [12,13].

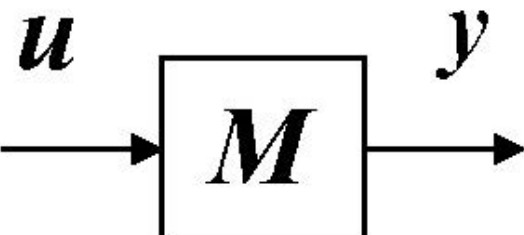

**Figure 1.** Linear system model $M$ block diagram representation with input $u$ and output $y$ (I-O data).

The component terms of the solution $\boldsymbol{y}(t) = C\boldsymbol{q}(t) = Ce^{Kt}\boldsymbol{q}(0)$ of the homogeneous equation $d\boldsymbol{q}/dt = K\boldsymbol{q}$, i.e., the response to initial conditions (ICs) $\boldsymbol{q}(0)$ (input $\boldsymbol{u} \equiv \boldsymbol{0}$) are the **modes** of model (1). This *fundamental set* of solution terms (modes) is $n$ linearly independent exponential functions of the form $t^k e^{\lambda_j t}$ ($k = 0, 1, \ldots, n - 1$ and $j = 1, 2, \ldots, n$), each satisfying $d\boldsymbol{q}/dt = K\boldsymbol{q}$. The $\lambda_j$ are *eigenvalues of K*. If they are distinct, the $n$ modes are $y_1 = e^{\lambda_1 t}$, $\ldots$, $y_n = e^{\lambda_n t}$. If they are repeated, then for each $\lambda_i$ of multiplicity $n_i$ (i.e., $n_i$ is the number of $\lambda_i$), there are $n_i$ modes: $e^{\lambda_i t}$, $t e^{\lambda_i t}$, $\ldots$, $t^{n_i - 1} e^{\lambda_i t}$, also totaling $n$.

Importantly, $n$ is the minimum dynamical dimension of model (1), i.e., the model cannot be reduced to a smaller number of ODEs that completely characterize dynamical responses $\boldsymbol{y}(t) = C\boldsymbol{q}(t) = Ce^{Kt}\boldsymbol{q}(0)$. This is paramount in tying the model with I-O data, as we see below. The eigenvalues $\lambda_j$ can be real or complex numbers. When they are complex, the modes are *oscillatory*.

We now give all of this a geometric interpretation to tie it to the discussion of nonlinear (NL) models and modes in Sections 10 and 11. Each mode actually has a **shape**, represented by an *eigenvector* component $v_j$, associated with the **frequency** (eigenvalue) component $\lambda_j$, so it can be represented geometrically as a **two-dimensional** function $c_j v_j t^k e^{\lambda_j t}$ each with its own eigenvalue $\lambda_j$ and ($n$-dimensional) eigenvector $v_j$. The intrinsic parameters

of $\boldsymbol{y}(t) = C\boldsymbol{q}(t) = Ce^{Kt}\boldsymbol{q}(0)$ are then the eigenvalues and eigenvectors of $K$, i.e., a set of $2n$ *structural invariants* of $K$. **Structural invariants** are one of many smallest sets of parameters completely determining or characterizing model dynamics [14].

*Example 1: **Simplest Case—Distinct Eigenvalues***

The dynamical shape of the solution modes, for recognition purposes as well as math analysis, depends in part on relationships among the $n$ eigenvalues $\lambda_1, \lambda_2, \ldots, \lambda_n$ of system matrix $K$. If they are *distinct* (none are repeated) and the corresponding eigenvectors are $v_1$, $v_2, \ldots, v_n$, the general solution of $d\boldsymbol{q}/dt = K\boldsymbol{q}$ can be written:

$$\boldsymbol{q}(t) = \sum_{i=1}^{n} c_i \boldsymbol{v}_i e^{\lambda_i t} \tag{2}$$

The $c_i$ are chosen to match the *initial conditions* (ICs) on $\boldsymbol{q}(t)$. For example, if only $q_1(0) \neq 0$ and all other ICs are zero, then Equation (2) gives: $\boldsymbol{q}(t) = [q_1(0)\sum_{i=1}^{n} v_1 e^{\lambda_i t} \ 0 \ \cdots \ 0]^T$. If the eigenvalues are not distinct, the solution is more complex but still multiexponential, and each repeated eigenvalue has its own mode, e.g., $c_1 e^{\lambda t}$, $c_2 t e^{\lambda t}$ are two different modes for repeated eigenvalue $\lambda$. Multicompartmental (MC) models can have repeated and/or complex eigenvalues in their responses.

We can generalize this for nonhomogeneous system inputs $\boldsymbol{u}(t) \neq \mathbf{0}$ in Equation (2). Since the exponential mode response to ICs is the same as the response to impulse inputs, i.e., $Bu(t) = q(0)\delta(t)$ in Equation (2), the system response to step, ramp, exponential and other (different) inputs—by linearity—also will be a sum of exponential modes and additional terms inherited from $u(t)$.

A most important feature of linear modes is that they are a **minimal set of invariants** of the dynamical system, and, for any set of ICs or input(s), the modes accurately capture essential system dynamics. Among other things, this means that linear ODEs of orders higher than $n$ can be accurately reduced (simplified) to order $n$ if the ODE has only $n$ modes, and this simplified model can be represented by a set of $n$ modes, each represented geometrically as a 2-dimensional function of the eigenvalue and eigenvector associated with each model. Whether all these modes are *visible* or *hidden* in data is the question we address next—the key to unraveling structure from data and designing new input–output (I-O) experiments that reveal more.

### 3. Modes as Data Signatures

As implied above, both *visible* and *hidden* modes play important roles in understanding dynamical signatures in data and thus the structure of systems from which they emanate. Using a possibly familiar analogy with large-scale data analysis in bioinformatics or computational statistics, finding the modes in dynamic system data is akin to finding the *principal components* (principal component analysis—PCA) or *singular values* (singular-value decomposition—SVD), of a large data matrix. This means visualizing a multivariate dataset—typically expressed in a high-dimensional data space, from its "most informative" viewpoint—its principal components—in a lower-dimensional space. Distinct *modes of chords and scales* in music are another albeit less mathematical example, with each mode sounding different but having a different arrangement of the same notes played in particular orders.

This is important because we can use these notions for developing dynamic system model *structure* and establishing the *dimensionality of the model observable in data*, in terms of *modes* associated with $n$ exponential solutions in the data. In brief, *visible modes*, in particular I-O data, collected from a system are visibly associated or identified with visible state variables in its model. *Hidden modes* in the data are associated with hidden state variables (hidden modes (state variables, compartments) are directly analogous to hidden states in a Markov model (*hidden Markov model*)), i.e., those that have no signature in the particular I-O data. They may be in the system, but they are not visible in that dataset.

Different experiments (input–output port configurations) are needed to discover them in new data.

To tie these relationships to the model *structure*, in particular multicompartmental structural models, we need some definitions.

## 4. State Variables, Compartments, Directed, Graphs, Pools and Species

We formalize and further clarify our primary nomenclature, paraphrasing classical definitions collectively in the contextual domain of biological systems, e.g., see [12,15,16].

A **compartment** is an amount of material *X* that acts kinetically (within a dynamic system) in a *homogeneously distinct* manner. Homogeneously distinct are the keywords here. This means entities within a compartment are indistinguishable from each other, they are "mixed completely". If they are not, they comprise more than one compartment. A compartment is **open** if it "leaks" into the environment; it is **closed** if it does not (Figure 2). Most (not all) compartmental models of real systems have at least one open compartment.

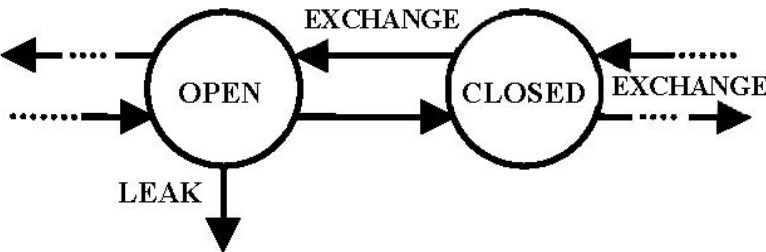

**Figure 2.** Two-compartment model representation (linear or nonlinear) illustrating one OPEN compartment, with a LEAK and a CLOSED compartment (no LEAK) exchanging material (solid arrows) with each other and possibly other compartments not shown (dotted arrows).

Every (physical) space containing compartments has a presumed volume in which the material substance (e.g., chemical "species", etc.) is homogeneously distributed. This **compartment volume** $V_i$ might be measurably real, or it might be virtual—called an *equivalent volume.*

*In other words*, in chemical, biological or other physical spaces, a compartment is an idealized store of a substance, characterized by its environment, physiochemical properties or both. All of the substance in a *given form* qualifies as a compartment, e.g., *active* or *inactive* or *receptor-bound* or *unbound* form—one compartment for each—because they generally have different dynamics. Alternatively, all of the substance in a homogeneously distributed (well-mixed) location, or all of it in a given form and location, qualifies as compartments, e.g., the mass of unbound drug $D_F$ (or hormone, lipid or ion) in blood or the mass of receptor-bound drug $D_R$ in blood or either in the liver (or another organ or organelle). Similarly, the total drug mass $D_F + D_R$ in the blood or liver (or another organ or organelle), is each a valid compartment in principle. Similarly, in modeling epidemiological systems, *Susceptible*, *Exposed*, *Infected* and *Recovered* (*SEIR*) populations are typically each considered valid, homogeneous compartments, as in COVID-19 modeling [17].

A **multicompartmental (MC) model** consists of a directed graph of two or more compartments interconnected, so there can be an exchange of material or change of state (into two or more compartments) among some or all of them and possible **leaks** from any of them (i.e., Figure 2). **Exchange** may occur by material transgressing some physical barrier (e.g., by diffusion or membrane transport) or by undergoing some physical or chemical transformation (e.g., metabolism, inactive to active state, etc.). Each **compartment** is represented by a single **state variable** in a multicompartmental (MC) model. Changes in the location of material generate different compartments (different state variables) for the same material, typically with different volumes. Changes in material form (e.g., precursor to product, carbohydrate to metabolite, infected to recovered, etc.) generate different compartments (different state variables)—possibly in the same "space" or "container" and with the same volume, and so on.

**Circles** are commonly used to represent compartments **arrows** to represent unidirectional transfers—which include inputs, influxes and effluxes (leaks)—and **dashed lines**, with a small open-circle endpoint, denote measured outputs (Figure 3). **Directed graphs** (**digraphs**) also are used to represent MC models, with **nodes** (vertices) representing compartments and **edges** as the arrows interconnecting compartments (nodes) or the environment, as in Figure 3. For linear systems, arrows and edges are typically represented by constant parameters (e.g., fractional transfer rates $k_{ij}$). For nonlinear systems, arrows and edges designate variable functions of constant parameters (e.g., $A$, $B$) and state variables $q_i$ (e.g., saturable functions, like $A/(B + q_i)$ [7].

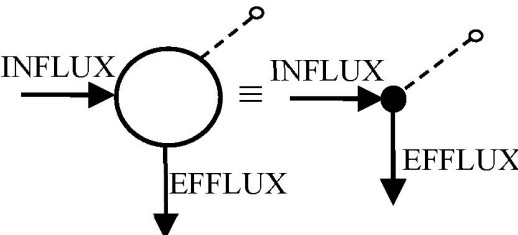

**Figure 3.** 1-compartment model and its directed graph equivalent.

A **pool** is the total amount of a substance in a system or subsystem [18], defined purposely loosely to suit the variety of ways different investigators were using it. We are a bit more precise here, defining **pool**—similarly, as a mass or volume size measure for a real or abstract space or material, emphasizing that it is not necessarily uniformly (homogeneously) distributed [19]. A compartment may be a pool, but not necessarily vice versa. On the other hand, we can speak of the "pool of *X* in compartments *i* and *j*" or the "pool of *X* in the whole organism" (or "whole compartmental model or system") because the term pool circumvents the homogeneity requirement on any of these entities. *Pool Models* are particularly useful for steady-state *flux balance analysis* (FBA) [6].

**Species** is a term commonly used in chemistry and molecular biology, usually defined as distinct molecular entities sharing the same chemical properties, and having the same chemical properties effectively means homogeneous. In a chemical or biological system kinetic (dynamical) model, distinct species are thus represented as distinct state variables—the classical definition of (homogeneous) compartment in a dynamic system model. Different definitions for some of these terms are embodied in the latest version (Level 3) of the *Systems Biology Markup Language SBML*, a "*bridge* language (*lingua franca*) for communication or exchange of mathematical models across programming tools" (*sbml.org*), with potential for confusion in modeling biological systems. In SBML, a "compartment is a well-stirred container of finite size where species may be located". And "species is a pool of entities of the same kind located in a compartment and participating in reactions (processes)". The main distinction of SBML from classical definitions (which we have adopted here) is that SBML "compartments" are *containers* of species, whereas classical compartments are unique homogeneous entities represented as state variables. The different meanings of *species*, *compartment* and *pool*—taken together—generate potential ambiguities for modeling.

A multicompartmental (MC) model can have any number of compartments. But its observable (visible) dimensionality is limited by the locations of inputs and outputs and thus the particular (noise-free) data available. This means the granularity—or extent—of multicompartmentalization is experiment-dependent. The minimum dimension—the smallest number of possible compartments—is equal to the number of its distinct *modes* visible in its output response for any particular specification of input and output locations.

## 5. Modes, Compartments and Data: Modes in Data = Minimum Compartment Number

In this and the following sections, we begin to show how modal analysis can be used to characterize (build) a system dynamical structure from temporal kinetic input–output (I-O) data collected from accessible I-O ports. This methodology is summarily unified in

Section 9. We have seen above that the modes of this data are the fundamental solutions of the ODEs representing the model, one for each of the state variables; and any zero-input response—i.e., response due only to ICs—is a linear combination of these modes. Thus, for linear multicompartmental models, the number of modes is equal to the number of state variables, which means **mode-count ≡ minimum compartment-count**.

The actual number of system compartments—identified as such or not—is typically greater than the minimum needed to represent the dynamics, in many cases very much larger than this minimum number. Probed compartment(s) are likely included in the count, but—at least in principle—additional "hidden" and unprobed compartments need to be discovered—typically by probing different I-O ports—to establish actual dynamical dimensionality, as illustrated below. This can be challenging for the modeler, no matter what the goals are.

### 6. Finding Modes (Compartments) Visible in Output Data Signals through Graphical Inspection

Not all modes and compartments (state variables) they represent are visible in a particular output $y(t)$, in response to a particular input $u(t)$—i.e., an input–output port pair. But they may be visible from other I-O data ports. We take advantage of the fact that modes are intrinsically responses to initial conditions (ICs) or, equivalently, *impulse* inputs. The procedure is to systematically test each I-O port pair of a *candidate model structure*, counting the number of modes in each designated output compartment in response to an impulse input into the designated input compartment. Candidate model structures are derived from available mechanistic biological system domain knowledge. *Alternative* candidates typically account for unknown or incomplete knowledge of compartmental connectivities. Checking all possible compartment I-O pairs in this manner then gives all possible modes in the system, i.e., all possible compartments in the candidate model. This, incidentally, reveals new experiment designs for finding otherwise hidden compartments.

Analysis can be performed mathematically, but it is easier through systematic inspection of the compartmental model graph. This graphical approach exposes visually why the full dimensionality of a model and the system it represents is difficult to establish. In the process, it also provides a more facile tool for discovering the hidden structure in the underlying system. The following two simple examples illustrate the procedure step-by-step. Additional, more complex examples can be found in [6].

*Example 2*: **2-Compartment Candidate Model with One Input and Two Outputs**

In the simple model of Figure 4, we assume zero ICs and a single impulse input. We relax these assumptions, later, to discover how ICs and other inputs affect results. The $k_{ij}$s are non-negative rate constants (time$^{-1}$), and $V_i$s are volumes. We are interested in the number of modes visible in outputs 1 and 2 under various conditions. We evaluate this number first mathematically and then more simply using graphical inspection.

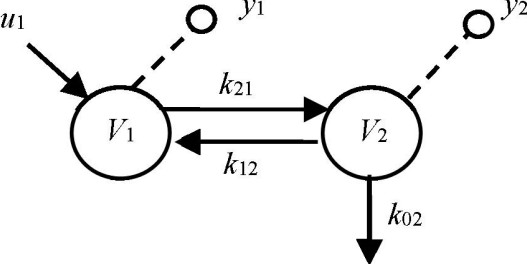

**Figure 4.** Example 2-compartment model, with one input and two outputs, for illustrating visible mode identification mathematically and by visual inspection.

*The Math Solution*: It can be shown [6] that both outputs are a sum of two real and distinct exponential modes:

$$y_1(t) = \frac{1}{V_1(\lambda_1 - \lambda_2)}\left[(k_{02} + k_{12} + \lambda_1)e^{\lambda_1 t} - (k_{02} + k_{12} + \lambda_2)e^{\lambda_2 t}\right], \quad y_2(t) = \frac{k_{21}}{V_1(\lambda_1 - \lambda_2)}\left(e^{\lambda_1 t} - e^{\lambda_2 t}\right)$$

and the two exponents (eigenvalues) are:

$$\lambda_{1,2} = \left(-(k_{02} + k_{12} + k_{21}) \pm \sqrt{(k_{02} + k_{12} + k_{21})^2 - 4k_{02}k_{21}}\right)/2 \le 0 \text{ for all } k_{ij} \ge 0$$

We have exactly two distinct (exponential) modes measurable in each compartment for an impulse input for all $k_{ij} > 0$. Now suppose $k_{21} = 0$ (and ICs remain zero), then $\lambda_2 = -(k_{02} + k_{12})$, which nulls $\lambda_1$ and the second exponential term in the equation for $y_1$, i.e., the solution for $y_1$ reduces to only *one mode*, and that for $y_2$ reduces to *zero modes* because the coefficient goes to zero (by L'Hopital's rule). Thus, the model with $k_{21} = 0$ will exhibit one mode in 1 and none in 2. That is what the math tells us.

*The Simpler Graphical Solution*: We redo the problem by reasoning physically about each case, $k_{21}$ nonzero and $k_{21}$ zero, i.e., with and without the forward connection from compartment 1 to 2. The principle here is that each compartment, when it is stimulated by a signal entering it, elicits a distinct mode associated with that compartment in its dynamic response. Whether this effect is visible in an output depends on the locations of the I-O probes and the compartment interconnections.

An input signal entering compartment 1 (an impulse input in this example) will first stimulate the first mode represented by that state variable—dynamic system compartment 1 itself. Then, if $k_{21}$ is nonzero, the signal travels to 2 via $k_{21}$, where it stimulates the second distinct mode, represented by that state variable—dynamic system compartment 2. Finally, with $k_{12}$ nonzero, material travels back to 1 via $k_{12}$, with a dynamic reflecting both the stimulus from 1 along $k_{21}$ *and* the corresponding dynamic due to stimulation of the distinct mode in 2. So we see *two* modes in both compartments. They go back and forth, feeding their signals forward and back, because they are connected reversibly.

*Other Parameterizations:* Now, using the same reasoning and inspection of the graph, we see that if $k_{12} = 0$ and $k_{21}$ is nonzero, we find only one mode in compartments 1 and 2 in compartment 2, for the same input and ICs, because the mode associated with compartment 2 is not visible in 1 because of the lack of connection back to 1 ($k_{12} = 0$). We discovered this using the mathematics above, but this graphical solution is easier.

*Nonzero ICs*: Going one step further, if we started with *zero* input, $k_{12} \ne 0$, $k_{21} \ne 0$ and nonzero ICs in both compartments, both would have *two* modes visible in them because the ICs are equivalent to impulse inputs. And, unlike the zero IC case above, when $k_{21} = 0$, compartment 1 would still show 2 modes because mode 2 would be stimulated by the nonzero IC in 2 and travel to 1 along $k_{12}$.

So nonzero ICs can be as important as inputs in establishing model complexity because—in essence—they are equivalent to impulse inputs and generate the same modal responses.

*Example 3*: **3-Compartment Candidate Model**

We add a third compartment, the only one measured in this example, and evaluate the number of modes visible in the output of compartment 3 in Figure 5, using only graphical reasoning. We again assume zero ICs everywhere, an impulse input and all $k_{ij} > 0$. In the previous example, we established that two modes would be visible in compartment 2 if it were measured. Despite the fact that compartment 2 is not measured in this problem, there are still two "excited" modes in 2, and they travel to compartment 3 along nonzero path $k_{32}$. So, at least two modes are visible in compartment 3 measurements. But, compartment 3 is itself a dynamic state variable compartment, and when it is stimulated, it generates its own mode. Stimulated by two modes coming from 2, it therefore exhibits *three* modes in its output. No math needed!

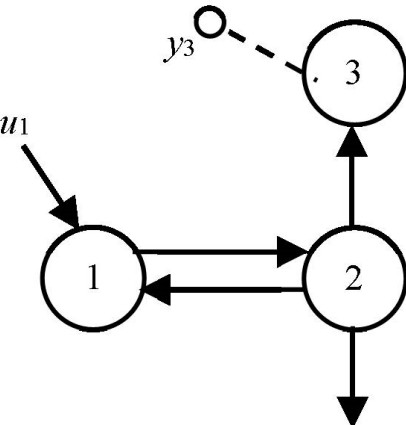

**Figure 5.** Mode determination by visual inspection for a more complex 3-compartment model.

*Different Inputs*: Going one step further, if the input $u_1$ were an *exponential function* distinct from the fundamental modes of the model, e.g., a single exponential input, then *four* modes would be visible at $y_3$ because the input "mode" would also be visible in the output. Inputs are external to the model; therefore, exponential inputs can be thought of as the output of a compartment external to the model (another mode!) and so on...

*Remarks:* In principle, the results above do not depend on parameter values, only structure, assuming all $k_{ij} > 0$ in the models. In reality, the relative magnitudes of the parameter values matter because depending on parameter values, some modes may be very small and difficult to detect in real data. This is a problem for model parameter sensitivity analysis, beyond our scope here.

Impulse inputs (approximate brief-pulse inputs in practice) are in principle maximally informative for modal analysis. If the input were instead a step, or ramp, or any integral of the impulse, the modal responses would be qualitatively the same—still exponential, with the same frequency $\lambda_i$ (eigenvalues), but with different shapes (eigenvectors). Arbitrary inputs also elicit the same, albeit less obvious, modal responses, and they would be more difficult to unravel. Nevertheless, they too can be accommodated if needed by using suitable (e.g., Fourier transform) analysis—also beyond our scope in this paper.

## 7. Automated Mode Detection Using Graph Theory Algorithms

The graphical mode-finding algorithm of the last section is readily automated using computer science graph theory tools. Either **Breadth-First Search (BFS) or Depth-First Search (DFS)** [20] can be used to count compartments with a directed path between any measurable compartments, i.e., those *reachable* from perturbed compartments, i.e., those receiving inputs or having nonzero initial conditions (ICs are the same as equivalent impulse inputs). Directed connectedness and the permutations of paths between inputs to different compartments and follow-up compartments are key to counting modes. **Pseudocode** for the imbedded mode-counting algorithm of the last section is given in Appendix A. This algorithm was originally written by Farhad Hormozdiari and augmented by Teaching Assistant Long Nguyen, both Ph.D. students at UCLA at the time. Notation $G(V,E)$ means a graph with $V$ vertices (nodes and compartments) and $E$ edges (rate-constant parameters).

## 8. Hidden Modes and Model Simplification

Hidden modes and compartments also have an upside in the context of model simplification. For state variable models, model **reduction (aggregation)** means finding a simpler representation (with fewer state variables) appropriate or useful for some purpose. Transfer functions (TFs) $H_{ij}(s)$ between pairs of inputs and outputs ($u_j$ and $y_i$) of multicompartmental models $\dot{q} = Kq + Bu, y = Cq$ are particularly convenient for this because *pole-zero cancellations* in $H_{ij}(s)$ reduce model dimensionality and therefore the state variable and compartment number between these I-O pairs. This means that a multicompartmental

model has hidden compartments if its TF has at least one pole-zero cancellation. Conversely, it has no hidden compartments if there are no pole-zero cancellations [21]. (We remark here that multicompartmental models also can have hidden oscillations (**hidden oscillatory modes**) in some compartments with complex eigenvalues (modes) hidden from outputs measured in a central compartment. This interesting phenomenon can occur in potentially physiologically realizable systems, modeled with *generalized mammillary models*, as developed in [21].

*Example 4*: **3- Compartment Candidate Model Transfer Function and Simplifications**

The I-O TF for the model of Example 3, Figure 5, is computed by using Laplace transform analysis as:

$$H_{31}(s) = \frac{k_{21}k_{32}}{s(s^2 + (k_{21} + k_{12} + k_{02} + k_{32})s + k_{21}(k_{02} + k_{32}))} \tag{3}$$

No pole-zero cancellation is possible in (3) with all $k_{ij} > 0$, so model dimensionality cannot be reduced from 3. If both the input and output are instead in compartment 2 here (TF $H_{22}$), the TF is:

$$H_{22}(s) = \frac{s(s + (k_{12} + k_{02} + k_{32}))}{s(s^2 + (k_{21} + k_{12} + k_{02} + k_{32})s + k_{21}(k_{02} + k_{32}))} \tag{4}$$

The zero and pole at the origin in (4) cancel each other, and the model is reduced to the second order. And if turnover rates in compartments 1 and 2 ($-k_{11} = k_{21}$ and $-k_{22} = k_{12} + k_{02} + k_{32}$) are equal, we still have a two-dimensional model:

$$H_{22}(s)|_{k_{21}=k_{12}+k_{02}+k_{32}} = \frac{s + k_{21}}{(s + k_{21})^2 - k_{21}k_{12}} \tag{5}$$

Although not the intention of this example, the number (two) of visible compartments in this $H_{22}$ example is available simply from the inspection of the graph: the mode generated in compartment 3 cannot be returned to and therefore be visible in 2 because there is no path from 3 to 2.

## 9. Deriving Mechanistic Models from Modal Analysis

The content of Sections 5–8 is summarized and consolidated here, delineating the process of characterizing (building) a system dynamical structure (as multicompartmental models) from dynamical input–output (I-O) data using modal analysis.

We assume at least some information/data is/are available about the biosystem *mechanism*, enough for formulating *alternative candidate* model structures with accessible or potentially accessible I-O ports. In biology, the system structure is usually partially known (the current knowledge base). To discover how the system can be more fully structured, we formulate *different hypotheses* about how it may be structured into different and explicit *candidate* models, with possibly unspecified *dimensionality* (number of state variable components or other complexity), and we test them against pertinent data (Examples 2 and 3 and Figures 4 and 5 in Section 6 are simple examples of two- and three-dimensional candidate models tested in this fashion, respectively). Notably, additional/alternative biomodel complexity is typically guided by the current knowledge base, often inferred by the known structure. Alternative candidates, for example, also can be based on open questions about its mechanistic structure as well as what the current literature or collaborators suggest.

Modal analysis directly answers the dimensionality question for a single set of I-O data (at least in principle). The modes of this data are the fundamental solutions of the ODEs representing the minimal model, one for each of the state variables *visible* in that dataset. For linear multicompartmental models, the number of modes is equal to the number of state variables, which means the mode-count $\equiv$ minimum compartment-count $\equiv$ minimal dynamical dimensionality of the model visualized from that I-O dataset.

Moving on, the actual number of compartments (state variables) is typically greater than the minimum needed to represent complete dynamics. Probed compartment(s) are included in the count, but—at least in principle—additional "hidden" and unprobed compartments need to be discovered to complete the model structure further. This is carried out as follows.

Modes intrinsically are responses to initial conditions (ICs) or, equivalently, *impulse* inputs. The procedure then is to systematically test each I-O port pair of a candidate model structure, counting the number of modes in each designated output compartment in response to an impulse input into the designated input compartment. Checking all possible compartment I-O pairs in this manner then gives all possible modes in the system, i.e., all possible compartments in the candidate model. This, incidentally, reveals new experiment designs for finding or otherwise exploring the dynamics of hidden compartments.

A procedure for computing alternative identifiable model candidates (indistinguishable I-O equivalent models), using a more sophisticated but mathematically equivalent methodology, can be found in [22], implemented in a freely accessible app at the website: http://biocyb1.cs.ucla.edu/DISTING/, (URL accessed on 13 July 2023).

## 10. Nonlinear Modes in Nonlinear Models

Nonlinear (NL) ODE models also have "transfer functions" (TFs), i.e., input–output solution response functions. Unlike linear models, however, their dynamic responses are dependent on inputs—input shapes or magnitudes. So we cannot write them only in terms of model parameters, like we do for linear model TFs. Nevertheless, the principle result above carries over to NL models. "Modes" in NL systems can cancel each other (like poles and zeros)—at least approximately—or be vanishingly small in magnitude, or present for vanishingly small transient times—thereby rendering complex structures less visible in data characterized by specific inputs and outputs. Similarly simplified NL models, also with input-dependent dynamics, are characteristic of enzyme kinetic models. We tie the linear mode concept together here with a novel view of *quasi-steady state approximations* (QSSA) in nonlinear biochemical kinetics models (described in Section 11)—viewed as *nonlinear modes* with the *same kinds of geometric properties* as linear modes. The goal is to better understand NL systems and facilitate NL model building and analysis.

The field of mechanics has a developed theory of nonlinear modes associated with certain oscillatory phenomena in mechanical systems. Biological systems share many of the same dynamical properties, not recognized as such in the field. We make the connection precise here.

As noted earlier, linear modes are a *minimal set of n-dimensional invariants* of the dynamical system and, for any set of ICs or input(s), the modes accurately capture essential system dynamics. Among other things, this means linear ODEs of an order higher than $n$ can be accurately reduced (simplified) to order $n$, and this simplified model can be represented by a set of *n two-dimensional modes* in a state vector space. Mode shape is represented by eigenvector component $v_j$, associated with a frequency (eigenvalue) component $\lambda_j$. The intrinsic parameters of the measurement model $\boldsymbol{y}(t) = C\boldsymbol{q}(t) = Ce^{Kt}\boldsymbol{q}(0)$ for the homogeneous linear system are then the eigenvalues and eigenvectors of $K$, a set of $2n$ structural invariants of $K$. Importantly, $\lambda_j$ can be complex as well as real.

In mechanics, modal analysis is typically concerned with vibrations and resonances in mechanical structures [23]. When their models are linear or NL and linearized about their equilibrium solutions—a common practice in analyzing stability properties—some $\lambda_j$ are complex. This means there exist oscillatory components in their dynamical behavior. In this context, linear modes are called **linear normal modes** (https://en.wikipedia.org/wiki/Normal_mode, accessed on 13 July 2023), or LNMs, and the governing equations of motion can be decoupled into LNMs, e.g., by diagonalization of their system matrix. In decoupled form, i.e., with a basis chosen to decouple the state variables, a linear system vibrates as if it consisted of independent oscillators governed by the (complex) modes, a property not shared with NL systems. Linear modal solutions also

obey the superposition principle, i.e., free and forced oscillations are conveniently expressed as linear combinations of individual LNM motions in phase space. Importantly, modal solutions have an *invariance* property, i.e., a motion initiated on one specific LNM—representable as a two-dimensional planar surface in phase space—has no effect on the remaining LNMs because they are decoupled. This property permits the modal concept to be readily extended to NL systems.

By analogy, for NL models of the form $dx/dt = f(x,u)$, a **nonlinear normal mode** (NNM) is defined similarly as a two-dimensional invariant manifold in phase space. Nonlinear systems, however, can exhibit highly complex behaviors not possible with linear systems and, for this reason, NNM analysis generally requires more complex mathematical handling [24]. (We remark here that the scope of this paper is limited to an important class of NL systems in biology and other physical systems with similar model structures and not to all NL systems. Hopefully, it will motivate more research focused on a wider variety of NL systems. For NL models, it conceptually introduces the bridge between NNM analysis in mechanics and nonlinear biosystem modeling and analysis presented in Section 11 via the NNM-QSSA analogy and invariant manifolds.).

Nonlinear modal analysis is about generating minimal reduced-order models that accurately capture the dynamics of higher-order models, with the help of NNMs. An effective and computationally efficient methodology for accomplishing this is based on the two-dimensional invariant manifold approach, as developed in [25], for example. Geometrically, (NL) NNMs—like LNMs—are represented by two-dimensional *nonplanar* surfaces in phase space. They are, however, tangent to the planar surfaces of their *linearized* ODE LNMs at the *equilibrium point* of their NL ODE system.

For mechanical system applications, the nonplanar surface NNM is parameterized by a single pair of state variables (displacement and velocity), chosen as master coordinates, with the remaining variables being functionally related to the chosen pair. In this special form, the original system behaves on the manifold surface like a nonlinear single-degree-of-freedom system. Solutions beginning in the manifold remain in it for all time, which extends the invariance property of LNMs to nonlinear systems. More general theory and applications of NNM analysis are available in [24].

## 11. Nonlinear Modes in Systems Biology

Conceptually, this treatment of NL normal modes in mechanics, as nonplanar two-dimensional manifolds in phase space, is directly extensible to a major class of NL systems biology models. The analogous application is reducing complex biosystem models using *conservation* relations, combined with *dynamical approximations* (quasi-steady state approximations, QSSA), common in biochemical reaction modeling. Enzymatic and cellular protein interaction networks (PIN) typically involve complex interactions among numerous cellular components, and their models typically involve sets of coupled first-order ODEs, each representing the state variable motions of the interacting cellular components. Dominant characteristics of these NL systems include numerous inter-variable conservation relations, which serve to *reduce system order*, and higher-order interactions that obey the *quasi-steady state approximation* (QSSA) [26] or total quasi-steady state (tQSSA) approximations [27,28]. Together, these constraints and approximations serve to simplify the dynamics of these systems substantially, with solutions on nonplanar two-dimensional manifolds in phase space, not unlike nonlinear normal modes in mechanics.

The mathematics of the classic Michaelis-Menten reaction of a single substrate *S* with a single enzyme *E* illustrates these constraints and QSSA (approximation) characteristics and is outlined in Appendix B. More complex biochemical interactions in cells follow the derivation of the M-M model.

In each of these interactions, the dominant dynamics are *second-order*, one each for the substrate and product, with all enzymatic reactions or other protein or metabolic interactions substituted by algebraic equations, based on the QSSA or tQSSA [6]. On a microscopic level, each interaction obeying one of these assumptions also has two nonlinear

modes, a fast one—on the "fast manifold"—and a slow one—on the "slow manifold". The fast one disappears nearly instantaneously (or very quickly) relative to the slow one—which represents the relatively longer-term behavior of the mechanisms that dominate overall biosystem dynamics. A single mode, the slow one, strongly dominates the dynamics and approximates the subsystem interaction reasonably well. For example, this behavior is illustrated in the phase space figures (with nullcline "manifolds") associated with a more complex four-dimensional Slyke–Cullen reaction feedback system model, where all trajectories starting at different points in phase space rapidly converge to a planar surface [29]).

This is a general extension of the QSSAs to higher-order biological systems and is directly analogous to model reduction of NNM analysis of invariant manifolds on the phase space of mechanical systems. This means NNM analysis is equivalent to QSSA analysis, which likely came first [26,30], with invariant manifolds of the biosystem in the substrate–product phase space of each reaction submodel.

## 12. Discussion

We have presented a cohesive viewpoint of the modal concept, merging classical linear with modern nonlinear dynamic system geometries. We used the classical multi-compartmental modeling paradigm for developing this viewpoint—with compartments representing state variables (nodes), material flux functions or constant parameters (edges) connecting them, and I-O ports explicitly stated. This is all in accord with the hypothesized, candidate real system structure and the experimental testing of it to discover its specific connectivity. This has been presented as a unifying paradigm for establishing minimal homeomorphic model structure from I-O data signatures—represented as modal responses on two-dimensional invariant manifolds—for NL as well as linear systems. Naturally encompassing these notions, compartmental modeling theory has a sound physical and mathematical basis, with much of it motivated early on by problems in chemical kinetics [31–33] Ironically, the new systems biology has had similar motivational forces—from biochemistry—but a few decades ago, the new theory separated itself from well-developed compartmentalization concepts. Developers of the Systems Biology Markup Language (SBML), a language designed for developing universally compatible software in biomodeling, ignored the past—likely inadvertently—in an important software "venue" receiving much attention in the systems biology modeling community at the time (http://en.wikipedia.org/wiki/SBML, accessed on 13 July 2023). They adopted an older and looser notion of compartment than that honed and refined by the established community of 20th-century compartmental analysts. Mones Berman, John Jacquez [15] and Sol Rubinow [13], for example, wrote books, ran national or university programs, or developed fairly widely used modeling software (SAAM, etc.) based on or about compartmental analysis methods—with compartments defined quite differently than in SBML lingo.

In the lingo of compartmental analysis, reviewed earlier, a compartment is an amount of material *X* that acts kinetically in a homogeneously distinct manner—ultimately a state variable in the language of dynamic systems and differential equations. In contrast, compartment in the "lingua franca" of SBML means a *space* that (typically) contains multiple state variables. This is *régressif* (pardon my French), taking us back to an earlier time in biology when physical spaces, like organs in a biological system, were conveniently called compartments. Colloquially speaking, they still are to some, but more likely they contain numerous sub-compartments in a myriad of forms—like nuclear, or cytosolic or tubule *X*, for example. As a consequence, both free and commercial software packages for systems biology modeling (SimBiology, Copasi, Cell Designer, etc.) are written or are being written with compartments that speak SBML, rather than classical modeling science, and many users (like myself) need an attitude adjustment as well as a workaround to use them.

Change is often useful, leading to progress, but in my view, not in this case. I have tried my best to show here that there is good reason to retain the classical notion of compartment in dynamic systems biology modeling, with direct application via modal

analysis for establishing system structure. In a nutshell, *compartment as a state variable* promotes the *congruence of relevant mechanistic model topology (connectedness) with modal I-O data*. Classical compartments circumscribe and embody the deep connection between signal data (measurements) and model structure—linear or nonlinear—at the same time leveraging well-developed tools of engineering and physics and compartmental modeling theory based on an analysis of the intrinsic properties of ordinary differential equations and their solution spaces.

## 13. Conclusions

The primary results of this work are summarized in the following points:

1.  Modes in system I-O responses accurately capture essential system dynamics for each I-O pair of a mechanistic (multicompartmental, linear or nonlinear) model, defining *minimal* system dimensionality (size and complexity) based on that I-O pair. In other words, the granularity of the model (or extent of its multicompartmentalization) is experiment-dependent, and the minimum dimension—the smallest number of possible compartments—is equal to the number of its distinct *modes* visible in its output response—for any particular specification of input and output (I-O) locations.
2.  More extensively, modal analysis methodology can establish more than model dimensionality from particular datasets. It also can help in designing new experiments to extract "hidden" structures and select among multiple *candidate* models (model distinguishability analysis) or simplify (reduce) models to their dynamical essentials.
3.  In particular, the testing of all I-O pairs of a hypothesized candidate multicompartmental structure can extract hidden modes and thus provide complete structure.
4.  Finding modes (and thus compartments) visible in output data signals by *graphical inspection* speeds up the model-building process and visually exposes new experiment designs for discovering compartment connectivity.
5.  By analogy, finding the modes in dynamic system data is akin to finding the *principal components* (principal component analysis—PCA), or *singular values* (singular-value decomposition—SVD), of a large dataset (e.g., bioinformatics data matrix), thereby simplifying it to the minimum dimension (size and complexity) for analysis.
6.  For linear systems, diagonalized modal solutions (linearized normal modes LNMs) have an *invariance* property, i.e., a motion initiated on one specific LNM—representable as a *two-dimensional planar surface in phase space*—has no effect on the remaining LNMs because they are decoupled. This property permits the modal concept to be readily extended to NL systems. A nonlinear normal mode (NNM) is a *two-dimensional invariant (nonplanar) manifold surface in phase space*, tangent to the planar surfaces of their *linearized* ODE LNMs at the equilibrium point of their NL ODE system.
7.  NL modal analysis is thus about generating *minimal reduced-order models* that accurately capture the dynamics of higher-order models in mechanics, with the help of NNMs.
8.  For biological systems, the analogous application is reducing complex biosystem models using *conservation* relations, combined with *dynamical approximations* (quasi-steady-state approximations, QSSA) common in biochemical reaction modeling. This means NNM analysis is equivalent to QSSA analysis, with invariant manifolds of the biosystem in the substrate–product phase space of each reaction submodel.
9.  Compartments **are** state variables, not spaces or containers of state variables.
10. Linear model simplification also can be helped by examining the poles and zeros of I-O transfer functions for cancellations.
11. Prospectively, this work introduces and explains the analogies between LNM and NNM analyses in mechanics and the QSSA in a broad class of regulated systems in biochemistry and systems biology. Hopefully, it will serve to promote the study of other NL system classes, like those that exhibit more complex behaviors.

**Funding:** This research received no external funding.

**Institutional Review Board Statement:** Not applicable.

**Informed Consent Statement:** Not applicable.

**Data Availability Statement:** Not applicable.

**Conflicts of Interest:** The author declares no conflict of interest.

## Appendix A

---

**Algorithm A1: Pseudocode for Calculating Mode Number for each Measurable Compartment.**

---

**Require:** *Graph G(V,E)*, node (compartment) *i* receives input
**Ensure:** Modes[*j*] mode-count for all compartments, for all *j* $\epsilon$ *V*
// Determine compartments reachable from the input
  **while** *v* $\epsilon$ *V*
  **if** v is reachable from i **then** // Using DFS or BFS:
*TrueMode*[*v*] ← 1
  **end if**
  **end while**
// Count modes based on connectivity:
**while** *v* $\epsilon$ *V* **do**
  **while** *v'* $\epsilon$ *V* **do**
  **if** *v* is reachable from *v'* **then** // *Using DFS or BFS:*
  **if** *TrueMode*[*v*] = 1 **then**
  *Modes*[*v*] ← Modes[*v*] + 1
  **end if**
  **end if**
  **end while**
**end while**

---

     This algorithm actually finds the maximum number of modes in each compartment because it is completely symbolic—with no regard for numerical values of model parameters. *Hidden* modes can be missed, e.g., those due to model parameters coinciding like the $k_{22} = k_{33}$ in Example 4.

## Appendix B. Full Michaelis–Menten (M-M) Dynamics Reduced to 2nd Order—A Very Common Paradigm in Biochemistry

     The following is adapted from [7]. The stoichiometric equation:

$$S + E \underset{k_{-1}}{\overset{k_1}{\rightleftharpoons}} ES \overset{k_2}{\rightarrow} P + E \tag{A1}$$

describes the Michaelis–Menten (M-M) reaction, in which enzyme *E* and substrate *S* initially react to reversibly form a *complex ES.* The complex then breaks down to form the *product P* plus *free* (not complexed) enzyme *E*. Let $c_S$, $c_E$, $c_{ES}$ and $c_P$ represent concentrations of the four molecular species (state variables) in (A1). Then, flux balance and/or the law of mass action applied to the three elementary reactions in (A1) leads directly to the following four coupled nonlinear ODEs:

$$\frac{dc_S}{dt} = -k_1 c_E c_S + k_{-1} c_{ES} \tag{A2}$$

$$\frac{dc_E}{dt} = -k_1 c_E c_S + k_{-1} c_{ES} + k_2 c_{ES} \tag{A3}$$

$$\frac{dc_{ES}}{dt} = k_1 c_E c_S - (k_{-1} + k_2) c_{ES} \tag{A4}$$

$$\frac{dc_P}{dt} = v_2 = k_2 c_{ES} \tag{A5}$$

The first three equations have a "hidden" redundancy and can be reduced to two ODEs, using an algebraic *conservation equation* derived simply by realizing that enzymes are present in the mix in either free *E* or complexed form *ES* and must always have a sum equal to the initial $c_E(0)$:

$$c_E(t) + c_{ES}(t) = c_E(0) \tag{A6}$$

for *t* > 0. Equation (A6) is then used to eliminate $c_E \equiv c_E(t)$ from the RHS of Equations (A2) and (A4). This gives:

$$\frac{dc_S}{dt} = -k_1 c_E(0) c_S + (k_1 c_S + k_{-1}) c_{ES} \tag{A7}$$

$$\frac{dc_{ES}}{dt} = k_1 c_E(0) c_S - (k_1 c_s + k_{-1} + k_2) c_{ES} \tag{A8}$$

The *quasi-steady state approximation (QSSA)* of M-M theory is based on the fact that equilibrium is very rapidly established among *E*, *S* and *ES* in accordance with the first (the reversible) reaction, virtually independent of the second reaction. Kinetically (and mathematically), this means that the change in complex *ES* concentration is *approximately* zero, relatively speaking, i.e., $\frac{dc_{ES}(t)}{dt} \cong 0$ for all $t > 0$. This then means that the LHS of Equation (A8) is approximately equal to zero, making it an *algebraic constraint equation* instead of a third ODE in the M-M model.

After some algebraic rearrangements of the equations, the reduced M-M ODE model has only *two* ODEs, combined in (A9), representing the (approximate) coupled nonlinear relationship between product *P* and substrate *S* concentrations in the reaction mix, parameterized by two constants, $v_{max}$ and $K_m$, in (A10).

$$\frac{dc_p}{dt} = -\frac{dc_S}{dt} \cong \frac{v_{\max}\, c_s(t)}{K_m + c_s(t)} \quad \textit{for all } t > 0 \quad \text{conc/time units} \tag{A9}$$

Parameters $v_{max}$, the **maximum value** of *v*, and $K_m$, the **Michaelis constant**, depend on the rate constants and *e*(0), i.e.,

$$v_{\max} = k_2 c_E(0) \quad \text{and} \quad K_m = \frac{k_{-1} + k_2}{k_1} \tag{A10}$$

The *necessary condition for validity of the QSSA is* $K_m + c_S(0) >> c_E(0)$ [26]. This condition is met (or approximately met) in most enzymatically regulated processes governing metabolism and protein–protein interactions in cells. The **total QSSA**, denoted **tQSSA** (derived similarly), is an even better approximation [27,28].

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
