# Peer review of "Linear and Nonlinear Modes and Data Signatures in Dynamic Systems Biology Models"

_applsci, doi:10.3390/app13179772_

Round 1

Reviewer 1 Report

This paper proposes how to constrain and analyze modal response data in delineating model structures to modal analysis for use in systems biology modeling to construct nonlinear and linear biological systems. It also implements a modal basis for the dynamic signature in the hypothetical input-output data associated with the structural model. This method helps in establishing the model dimensions, selecting multiple candidate models and simplifying the nature of the model. However, there are some problems: 

1. Please add detailed figure notes to each figure in the paper. 

2. The authors discuss in detail the case of different eigenvalues in the linear model, would the results still be the same if the model corresponded to the case of different eigenvalues?

3.Visible and hidden patterns each play an important role in understanding how the dynamics in the data are characterized?

4. How did the authors characterize the dynamic structure of the system from time-dynamic input-output data collected from accessible I-O ports?

5. The authors need a brief explanation of the principles of the quasi-steady state approximation (QSSA) in nonlinear biochemical kinetic modeling?  

6. Nonlinear systems can exhibit highly complex behaviors not possible with linear systems. These include modal interactions, saturation, limit loops, bifurcations, jumps, subharmonics, superharmonics and internal resonances, and chaos. Did the authors consider the presence of these factors in their analysis of nonlinear systems? If a nonlinear system can generate chaotic oscillations, how is its nonlinear modal analysis performed? 

The writing should be revised.

Author Response

Please see the attachment, which has the responses to both reviewers.

Reviewer 2 Report

This paper developed the modal basis for dynamical signatures in hypothetical (perfect) input-output data associated with a structural model and for nonlinear as well as linear biosystems. The paper should be interesting. Considering the following comments is recommended and may improve your manuscript:

-References should be from the Web of Science 2020-2023 (50% of all references, 30 references at least).

- What will society have from the paper?

-What is the result of the analysis?

-Text should be formatted. Especially given the number of each chapter and the number of each reference, or else readers would fail to understand the relation between each chapter clearly.

- Figures should have high quality and the figure's title should be further exhibited.

-Add some future scope. How can your work be expanded?

- The Introduction is chaotic, so the readers fail to capture the novelty and main analysis result of this paper quickly. Need to highlight the novelty of this paper and describe the current research status in detail.

-Add a conclusion and point out what have you done and what results were obtained one by one.

Author Response

Please see the attachment.  It has the replied to both reviewers.

Round 2

Reviewer 2 Report

  • All necessary modifications have been made.